# Effect of Intravenous Ketamine on Hypocranial Pressure Symptoms in Patients with Spinal Anesthetic Cesarean Sections: A Systematic Review and Meta-Analysis

**DOI:** 10.3390/jcm11144129

**Published:** 2022-07-16

**Authors:** Xiaoshen Liang, Xin Yang, Shuang Liang, Yu Zhang, Zhuofeng Ding, Qulian Guo, Changsheng Huang

**Affiliations:** 1Department of Anesthesiology, Xiangya Hospital Central South University, Changsha 410008, China; 198112351@csu.edu.cn (X.L.); doctorxinyang@hotmail.com (X.Y.); liangshuang@csu.edu.cn (S.L.); zhangy20210618@163.com (Y.Z.); dzfzhuofeng@163.com (Z.D.); qulianguo@hotmail.com (Q.G.); 2National Clinical Research Center for Geriatric Disorders, Xiangya Central South University, Changsha 410008, China

**Keywords:** cesarean section, ketamine, spinal anesthesia, intracranial hypotension, post-dural puncture headache

## Abstract

Background: Pregnant women are more likely to suffer post-puncture symptoms such as headaches and nausea due to the outflow of cerebrospinal fluid after spinal anesthesia. Because ketamine has the effect of raising intracranial pressure, it may be able to improve the symptoms of perioperative hypocranial pressure and effectively prevent the occurrence of hypocranial pressure-related side effects. Method: Keywords such as ketamine, cesarean section, and spinal anesthesia were searched in databases including Medline, Embase, Web of Science, and Cochrane from 1976 to 2021. Thirteen randomized controlled trials were selected for the meta-analysis. Results: A total of 12 randomized trials involving 2099 participants fulfilled the inclusion criteria. There was no significant association between ketamine and the risk of headaches compared to the placebo (RR = 1.12; 95% CI: 0.53, 2.35; *p* = 0.77; I² = 62%). There was no significant association between ketamine and nausea compared to the placebo (RR = 0.66; 95% CI: 0.40, 1.09; *p* = 0.10; I² = 57%). No significant associations between ketamine or the placebo and vomiting were found (RR = 0.94; 95% CI: 0.53, 1.67; *p* = 0.83; I² = 72%). Conclusion: Intravenous ketamine does not improve the symptoms caused by low intracranial pressure after spinal anesthesia in patients undergoing cesarean section.

## 1. Introduction

The proportion of cesarean sections in deliveries is increasing due to the gradual maturation of birth assistance techniques and social factors [1,2]. Cesarean delivery can help pregnant women who are not allowed to deliver normally due to problems such as pelvic stenosis, uterine opening insufficiency, and placenta previa, as well as avoiding the occurrence of pelvic floor injury, urinary incontinence, vaginal relaxation and other complications caused by natural labor [3,4]. However, weight gain during pregnancy leads to subcutaneous fat thickening and unclear exposure between vertebrae, which inevitably increases the risk of spinal anesthesia and post-puncture complications [5]. Hypocranial pressure caused by cerebrospinal fluid outflow is one of the common complications after spinal anesthesia, mainly manifested as headache after dural puncture [6], especially in terms of obstetric anesthesia [7]. In addition, low intracranial pressure can cause nausea and vomiting [8]. Ketamine is known as a non-competitive, high-affinity N-methyl-D-aspartic acid (NMDA) receptor antagonist. Unlike sedative drugs such as propofol and etomidate, it also has a good analgesic effect, which has led to its common use in postoperative analgesia in patients undergoing surgery [9]. Another characteristic of ketamine is that it can increase the intracranial pressure [10,11], and by this compensatory effect, it can relieve headaches caused by low intracranial pressure [12]. However, evidence from other studies is inconsistent regarding whether ketamine reduces the incidence of headaches in patients undergoing cesarean delivery. Therefore, as evidence accumulated, we integrated relevant randomized controlled trials (RCTs) and performed a systematic review and meta-analysis to assess whether ketamine reduced the incidence of headaches in patients undergoing cesarean section under spinal anesthesia. Herein, we hypothesize that ketamine may be an effective agent for the prevention of hypocranial pressure syndrome. We hope to provide evidence for the value of ketamine in the prevention of headaches caused by low cranial pressure.

## 2. Materials and Methods

### 2.1. Search Strategy and Selection Criteria

This study was designed according to the Preferred Reporting Items for Systematic Reviews and Meta-Analyses (PRISMA) guidelines. We searched the Medline, Embase, Web of Science, and Cochrane databases from 1976 to 2021 (search strategies are detailed in Appendix A), resulting in the inclusion of 2099 patients. The search policy was designed and executed by two authors (Xing, Yang, and Shuang, Liang).

The inclusion criteria were as follows: (a) randomized controlled trial; (b) American Society of Anesthesiologists (ASA) I–II; (c) patient undergoing cesarean section with spinal anesthesia; (d) participant treated intravenously with ketamine; (e) article written in English. The exclusion criteria were as follows: (a) unpublished clinical trial; (b) full text unavailable; (c) control group received a drug other than normal saline.

### 2.2. Data Extraction and Quality Assessment

The data included in each study were independently extracted by two researchers and summarized in an Excel spreadsheet (Xiaoshen Liang and Yu Zhang). We extracted the following information from these studies: the first author’s name, year of publication, number of patients, age, BMI (height and weight values were provided in some studies), ketamine dose, needle size, relevant outcomes (the incidence of perioperative headaches, nausea, and vomiting), and reported results. This analysis assessed the outcomes, including the incidence of perioperative headaches, nausea, and vomiting, which are clinical manifestations of low cranial pressure (Table 1). Two other investigators (Xing, Yang, and Shuang, Liang) independently evaluated the included studies according to the Cochrane Handbook for Systematic Reviews of Interventions 5.0. The specific contents of the assessment’s “risk of bias” table included the following: adequate sequence generation, allocation of concealment, blinding, incomplete outcome data, free of selective reporting, and free of other bias.

### 2.3. Statistical Analysis

We evaluated the dichotomous data using risk ratios with 95% confidence intervals. Heterogeneity is reported using I² statistics; I² > 50% indicates significant heterogeneity. When there was significant heterogeneity between the two included studies (*p* < 0.05 or I² > 50%), the size of the combined effect was calculated using the random-effects model; otherwise, the fixed-effects model was used. If significant heterogeneity existed, we omitted a study and instead looked for potential sources of heterogeneity. We also performed heterogeneity analysis by subgroup analysis. A leave-one-out test, consisting of calculating the pooled risk ratio by sequentially excluding one study, was performed to identify studies with a strong influence on the results. The publication bias was assessed by using funnel plots. Review Manager Version 5.3 (The Cochrane Collaboration, Software Update, Oxford, UK) was used to perform the meta-analyses. *p*-Values <0.05 were considered statistically significant.

## 3. Results

### 3.1. Eligible Studies and the Characteristics

By searching the databases mentioned above, 208 studies were retrieved for the systematic review. Eighty-seven duplicate studies were excluded. After screening the titles and abstracts, only thirty studies were considered eligible for full reading. Twelve studies met all the selection criteria [12,13,14,15,16,17,18,19,20,21,22,23] as determined through browsing the titles, abstracts, and full texts (Figure 1). The included articles were published between 2005 and 2020, with sample sizes ranging from 28 to 163. All of the patients were delivered by cesarean section under spinal anesthesia and injected with ketamine intravenously at 0.15–0.5 mg/kg. The baseline characteristics of all the studies were statistically similar. The characteristics of the included randomized controlled trials are summarized below (Table 1).

Of all the included studies, three documented the incidence of postoperative headaches at different time points; the remaining studies did not explain the specific time points of occurrence. Therefore, the number with the highest incidence in these studies was chosen as the overall incidence figure based on the authors’ description of the recording method. One study used two doses of ketamine compared to a placebo group, so we combined the two into one group when extracting data. In all the included articles, we could only calculate the respective totals of postoperative headaches, nausea, and vomiting in each study, as most articles did not include headaches, nausea, or vomiting as primary outcomes. Only one study [12] documented, in detail, the incidence of headaches occurring immediately postoperatively, 4 h postoperatively, 12 h postoperatively, and 24 h postoperatively. And in this study, a significant difference was found in the incidence of headaches between the ketamine and control groups in the immediate postoperative period and 4 h postoperatively, while no significant difference was found in the 12-h and 24-h postoperative periods. However, in another study [18], who recorded the incidence of headaches at 5 min postoperatively, 15 min postoperatively, and at the time of leaving the operating room, the incidence of headaches was higher in the ketamine group than in the control group.

### 3.2. Quality Assessment of the Selected Studies

The Cochrane Quality Assessment Form indicated that the majority of the studies were regarded as “low risk” or “unclear risk”. The risk after consolidation is shown in Figure 2.

### 3.3. Effect of Interventions

#### 3.3.1. Headaches

The incidence of postoperative headaches was assessed in this meta-analysis. Considering the significant heterogeneity (I² > 50%), we used the random-effects model to calculate the pooled effect size. There was no significant improvement in the ketamine group compared to the placebo (RR = 1.12; 95% CI: 0.53, 2.35; *p* = 0.77; I² = 62%) (Figure 3).

#### 3.3.2. Nausea

The incidence of postoperative nausea was assessed in this meta-analysis. We used the random-effects model to calculate the pooled effect size. The nausea was not significantly different between the two groups (RR = 0.66; 95% CI: 0.40, 1.09; *p* = 0.10; I² = 57%) (Figure 4).

#### 3.3.3. Vomiting

The incidence of postoperative vomiting was assessed in this meta-analysis. We used the random-effects model to calculate the pooled effect size. Vomiting was not significantly different between the two groups (RR = 0.94; 95% CI: 0.53, 1.67; *p* = 0.83; I² = 72%) (Figure 5).

### 3.4. Subgroup Analysis

Considering that different durations of ketamine administration may be one of the important factors affecting the outcome indicators, we conducted a subgroup analysis of the included studies, which were divided into two subgroups: pre-delivery administration and post-delivery administration. However, the results of the three outcome measures did not change, and there was still no significant difference between the ketamine and control groups (Figure 6a–c).

### 3.5. Sensitivity Analysis and Publication Bias

In sensitivity analyses, we calculated the combined effect size by the consecutive exclusion of one study, but the indicators remained non-significantly different between the two groups, indicating that our primary estimates of headaches, nausea, and vomiting were robust. No obvious publication bias was found through the visual inspection of the funnel plots (Figure 7a–c).

## 4. Discussion

A cesarean section is one of the most common operations in the operating room, and spinal anesthesia can provide sufficient surgical anesthesia [24]. Among the many complications of spinal anesthesia, post-dural puncture headaches are one of the more common, usually due to cerebrospinal fluid flowing out of the patient’s dural puncture site [25,26]. Moreover, due to significant weight gain during pregnancy, most pregnant women have thickened subcutaneous fat, making the administration of spinal anesthesia more difficult, which also leads to an increased risk of cerebrospinal fluid outflow leading to hypocranial pressure [27,28]. Headaches are a characteristic symptom of low intracranial pressure [29], most importantly, postural headaches, accompanied by nausea and vomiting [30]. Therefore, we regard the incidence of headaches, nausea, and vomiting as an indicator for assessing the treatment of hypocranial pressure. Studies have shown that hypocranial pressure syndrome affects the prognosis and recovery of pregnant women [31]. Although different treatment modalities are available depending on the severity, including lying down, massive fluid administration, analgesic medications, and epidural blood patches [32,33], the effective prevention of PDPH is important.

Ketamine is a traditional intravenous anesthetic that not only has sympathetic excitatory effects [34], but also increases cerebral blood flow by dilating cerebral blood vessels, triggering an increase in intracranial pressure [35]. The mechanism by which ketamine increases intracranial pressure has been discerned, and it is noteworthy that ketamine can cause an increase in intracranial pressure by increasing the partial pressure of carbon dioxide and thus dilating the cerebral vasculature while preserving natural respiration [36]. Therefore, the intracranial pressure-raising effect of ketamine was allowed during spontaneous breathing in the cesarean section patients included in our study. Furthermore, although other studies have suggested that ketamine may cause a decrease in intracranial pressure [37,38] or have no significant effect on intracranial pressure [36,39,40], the prevailing view remains that ketamine has an increased effect on intracranial pressure. However, it is unclear whether this increase in intracranial pressure is sustained and whether it has associated adverse effects in the perioperative period.

In this meta-analysis, we demonstrate that intravenous ketamine does not improve headaches, nausea, and vomiting due to low cranial pressure in patients undergoing spinal anesthesia for cesarean delivery. Considering that the different time points of intravenous ketamine administration may be a key variable in this assessment, we performed a subgroup analysis with fetal delivery as the dividing line, but no difference between the ketamine and control groups was demonstrated. We concluded that the different time points of administration had no effect on the therapeutic effect of ketamine.

The main limitation of this study is that, for most of the included studies, we do not know whether the patients experienced cerebrospinal fluid leakage—this would affect our ability to accurately determine whether these patients had low cranial pressure.Thus, we could not distinguish whether the patients’ headaches, nausea, and vomiting were caused by low cranial pressure or the psychiatric-related side effects of ketamine [41,42]. Therefore, based on the available evidence, we believe that ketamine (range 0.15 mg/kg to 0.5 mg/kg) is not effective in preventing hypocranial-pressure-related symptoms in patients undergoing cesarean section under spinal anesthesia. However, esketamine, with its stronger analgesic effect and fewer psychiatric side effects, has gradually replaced ketamine in clinical anesthesia [43]. This drug may be an effective agent for the prevention of hypocranial pressure.

## Figures and Tables

**Figure 1 jcm-11-04129-f001:**
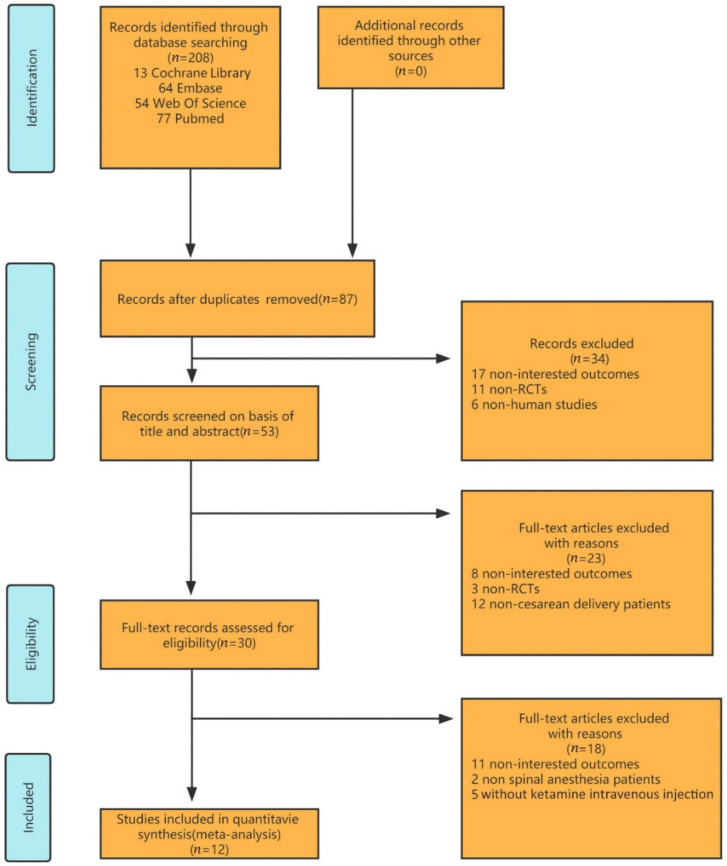
Flow diagram of the study selection.

**Figure 2 jcm-11-04129-f002:**
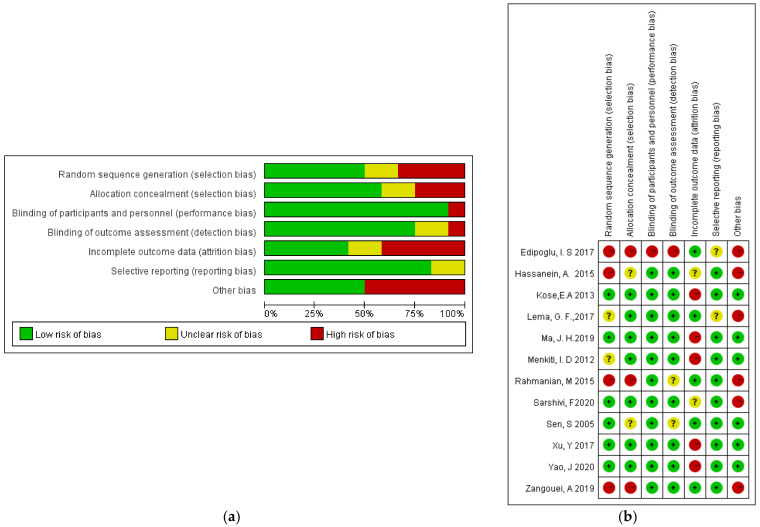
Risk of bias: (**a**) a summary table of the review authors’ judgements for each risk of bias item for each study; (**b**) a plot of the distribution of the review authors’ judgements across studies for each risk of bias item [12,13,14,15,16,17,18,19,20,21,22,23]. “+” Low risk of bias; “?” unclear risk of bias; “−” high risk of bias.

**Figure 3 jcm-11-04129-f003:**
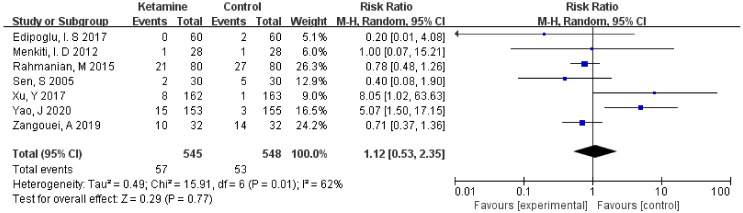
Forest plot of headaches [12,13,14,15,16,17,18].

**Figure 4 jcm-11-04129-f004:**
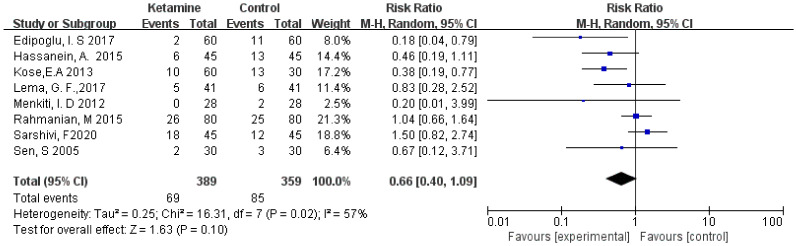
Forest plot of nausea [13,14,15,16,19,20,21,23].

**Figure 5 jcm-11-04129-f005:**
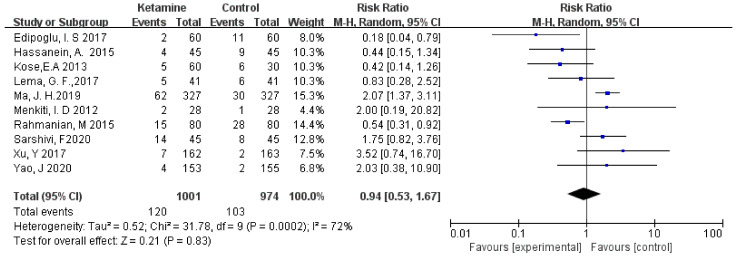
Forest plot of vomiting [14,15,16,17,18,19,20,21,22,23].

**Figure 6 jcm-11-04129-f006:**
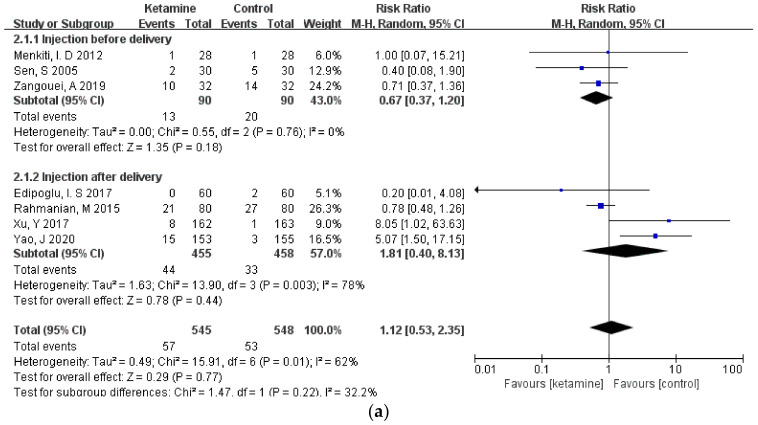
Forest plot of subgroups [12,13,14,15,16,17,18,19,20,21,22,23]: (**a**) headaches; (**b**) nausea; (**c**) vomiting.

**Figure 7 jcm-11-04129-f007:**
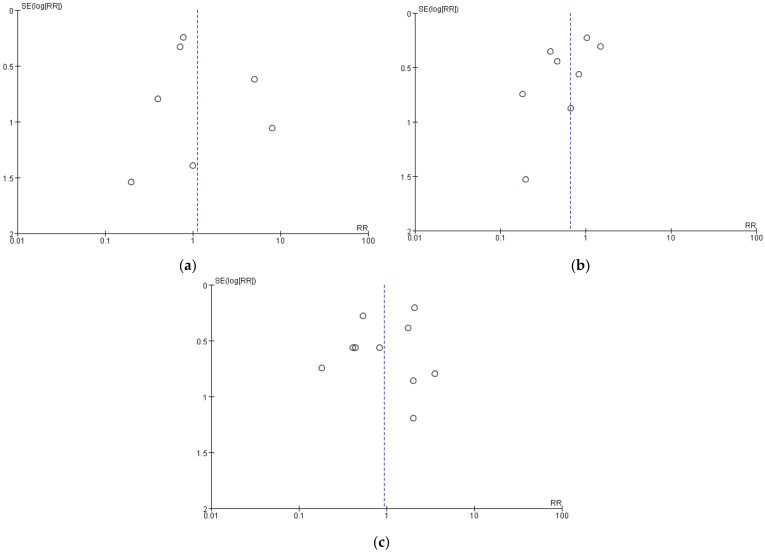
Funnel plot of comparison: (**a**) headaches; (**b**) nausea; (**c**) vomiting.

**Table 1 jcm-11-04129-t001:** Characteristics of the included studies.

First Author	Year	Number (K; C)	Age, Years (K; C)	BMI	Height, cm (K; C)	Weight, kg (K; C)	Surgery Time, min (K; C)	Ketamine Dose (mg/kg)	Needle Size (G)	Relevant Outcomes	Reported Results (K; C)
Sen, S. et al. [13].	2005	30; 30	26.3 ± 5.3; 27.1 ± 4.6	/	162 ± 6.1; 160.0 ± 5.2	78.1 ± 7.8; 73.2 ± 9.4	41.9 ± 9.2; 48.3 ± 6.4	0.15	25	Headache, nausea	1. Headache 2; 5 2. Nausea 2; 3
Menkiti, I. D. et al. [14].	2012	28; 28	30.3 ± 4.0; 29.8 ± 3.1	/	1.64 ± 0.03; 1.61 ± 0.12	73.7 ± 6.2; 72.2 ± 5.0	56.3 ± 8.6; 55.6 ± 8.6	0.15	26	Headaches, nausea, and vomiting	1. Headaches 1; 1 2. Nausea 0; 2 3. Vomiting 2; 1
Rahmanian, M. et al. [15].	2015	80; 80	27.4 ± 4.8; 27.6 ± 4.4	/	/	78.5 ± 11.9; 77.7 ± 10.8	40.2 ± 3.8; 39.8 ± 3.2	0.25	25	Headaches, nausea, and vomiting	1. Headaches 21; 27 2. Nausea 26; 25 3. Vomiting 15; 28
Edipoglu, I. S. et al. [16].	2017	60; 60	28.68 ± 5.82; 28.43 ± 4.82	33.54 ± 4.61; 33.85 ± 4.94	/	/	27.30 ± 7.36; 27.93 ± 5.34	0.15	26	Headaches, nausea, and vomiting	1. Headaches 0; 2 2. Nausea 2; 11 3. Vomiting 2; 11
Xu, Y. et al. [17].	2017	162; 163	31 ± 4; 32 ± 4	27 ± 3; 28 ± 3	/	/	43.8 ± 14.4; 44.0 ± 12.6	0.25	25	Headaches and vomiting	1. Headaches 8; 1 2. Vomiting 7; 2
Zangouei, A. et al. [12].	2019	32; 32	/	/	/	/	/	0.15	27	Headaches	1. Headaches 10; 14
Yao, J. et al. [18].	2020	153; 155	30 ± 4; 30 ± 3	29 ± 3; 28 ± 3	/	/	/	0.25	/	Headaches	1. Headache 15; 3
Kose, E. A. et al. [19].	2013	(30 + 30); 30	{28.2(18–43) + 26.8 (20–45)}; 27.3 (18–43)	{26.9 ± 5.9+ 26.3 ± 6.1}; 27.8 ± 6.8	/	/	68 ± 6; 71 ± 5; 65 ± 7	0.25 and 0.5	25	Nausea and vomiting	1. Nausea 10; 13 2. Vomiting 5; 6
Hassanein, A. et al. [20].	2015	45; 45	29.4 ± 7.2; 30 ± 6	/	163 ± 4; 162 ± 6.5	72 ± 9; 69 ± 13	/	0.4	25	Nausea and vomiting	1. Nausea 6; 13 2. Vomiting 4; 9
Lema, G. F. et al. [21].	2017	41; 41	26 (6); 26 (7)	/	160 (10); 160 (11)	60 (9); 60 (14)	50 (20); 45 (20)	0.2	22–25	Nausea and vomiting	1. Nausea 5; 6 2. Vomiting 5; 6
Ma, J. H. et al. [22].	2019	327; 327	/	27.5 ± 3.1; 29.4 ± 26.6	/	/	/	0.5	25	Vomiting	1. Vomiting 62; 30
Sarshivi, F. et al. [23].	2020	45; 45	30.8 (±4.5); 29.8 (±4.5)	29.4 ± 3.3; 29.1 ± 8.8	159.12 ± 4.52; 158.14 ± 4.22	68.19 ±5.48; 67.37 ± 8.12	51 (±18); 54 (±15)	0.3	25	Nausea and vomiting	1. Nausea 18; 12 2. Vomiting 14; 8

Table notes: K, ketamine group; C, control group; BMI, body mass index.

## Data Availability

No new data were created in this study. Data sharing is not applicable to this article.

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
