# Peer review of "Effect of Intravenous Ketamine on Hypocranial Pressure Symptoms in Patients with Spinal Anesthetic Cesarean Sections: A Systematic Review and Meta-Analysis"

_jcm, 2022, doi:10.3390/jcm11144129_

Round 1
Reviewer 1 Report
Thank you for submitting the manuscript. I read your paper with great interest and attention. The question you intend to solve with your paper is really interesting.Anyone who has been confronted with obstetric anesthesia knows that headache after spinal anesthesia is a frequent and difficult to treat complication. The idea of ketamine has its own rationale, and therefore the result of your meta-analysis is even more important. I ask you to harmonize the style of the abstract with that of the Journal.Furthermore, I ask you to modify line 167 as the caesarean section is not performed only in urgency, but has specific indications also in election. Furthermore, I ask you to specify if you have excluded articles that are not written in English.Apart from these small and simple revisions, I am convinced that your work is truly of excellent quality. I hope my comments will be useful to you.
Kind Regards
Author Response
Response: We are pleased that our study has been recognized and appreciated by this reviewer, which is very encouraging for us. We would also like to thank this reviewer for pointing out the critical points and shortcomings in our study. This has helped us a lot in our manuscript. The manuscript has been carefully and thoroughly edited, and we have revised the manuscript to make it clearer and easier to follow. We have made the following changes in response to this reviewer's valuable comments:
- We modified the format of the abstract to match the requirements of the journal to which we were submitting the manuscript.
- We then corrected the description in line 167 of the original manuscript because not all cesarean sections are performed under emergency surgery, so we revised this section to avoid misunderstanding by reviewers and readers. We now list the revised sentences as follows: “Cesarean section is one of the most common operations in operating room, and the spinal anesthesia can provide a sufficient analgesic”.
- In the inclusion criteria we developed, we did include only articles written in English. We refined the inclusion and exclusion criteria: “The Inclusion criteria were as follows: a. Randomized controlled trial, b. American Society of Anesthesiologists ( ASA ) Ⅰ-Ⅱ, c. Undergoing cesarean section with spinal anesthesia, d. participants treated intravenously with ketamine, e. Articles written in English. The exclusion criteria were as follows: a. Unpublished clinic trail; b. Unavaila-ble full texts; c. Control group was a drug other than normal saline”.
We also mark the modified text with red in the revised version.

Reviewer 2 Report
I would like to congratulate the authors on this paper evaluating the impact of ketamine on side effects following spinal anesthesia. Overall, the paper provides a good idea, but is difficult to understand at times. The incidence of PDPH following spinal anesthesia is very rare and the sample size reviewed in this paper are likely not powered to determine if ketamine impacted this. Was the use of ketamine for PDPH considered as the primary search? These patients with known headache or inadvertent dural puncture during epidural placement are likely more likely to have changes in symptoms with ketamine treatment compared to those who underwent spinal anesthesia alone for cesearean delivery.
-The use of the term "vomit" throughout the manuscript should be replaced with "vomiting" for most of the instances
-Page 1, Line 32: It is not clear to me what "progressive technology maturity" means.
-Page 1, Line 32: The sentence doesn't read clearly. Consider removing "because it can not only" and replace with "Cesarean delivery [help pregnant women...]"
-Page 2 line 59: you state you "screened 2099 patients" but do you mean 2099 manuscripts? If so, this should be included in the results. If not, please clarify what yoyu mean by screening patients.
-Page 2 Line 63: your study aims to evaluate patients undergoing spinal anesthesia, yet your inclusion criteria includes patients undergoing surgery with general anesthesia. Please provide clarification. Also include anesthesia types to be excluded in the exclusion criteria and procedure types/anesthesia types to be included.
-Were patients with known dural punctures included (such as inadvertent dural puncture during epidural placement)?
-Page 2 Line 70: "Relevant Outcomes" and "Reported Results" shouldn't be capitalized. You can list the outcomes within parentheses after "relevant outcomes" and remove the subsequent sentence.
-Page 2 Line 73: This sentence and the subsequent sentences discussing characteristics of studies evaluated should be moved to the results section. Please include the steps for review of the identified studies, but don't include individual information
-The methods section should not refer to figures. The results/discussion section should reference figures.
-The use of the term "literatures" should be changed to "studies" or "manuscripts".
-Table 1: the use of "K/C" makes the control group appear as the denominator for the ketamine group. Please correct this and provide relevant denominators for each characteristic reported.
-Table 1: Was information regarding spinal needle size or shape included in data collection? Both of these factors would contribute to the potential for PDPH.
-Page 10 Line 156: It is not clear to me why you abandoned one study each time in the sensitivity analysis. What was this sensitivity analysis aiming to accomplish? Please add this to the methods section and provide further context within the results.
-Page 10 Line 166. Was this paper limited to only emergency cesarean sections? This is the first mention of emergency cesarean delivery. Please reconsider the use of "emergency". Additionally, the spinal anesthetic is providing surgical anesthesia, not "analgesia".
-Page 10 Line 174. Consider removing "decubitus", using the term "analgesic medications" instead of "analgesic sedation", and "epidural blood patch" instead of "epidural blood filling therapy. Additionally, I don't quite understand what "It is more important for patients to effectively prevent and alleviate their perioperative discomfort" means in this context
-Page 10 Lines 176-180: It is tough to follow the wording of these sections. Please make more clear.
-Page 10 Line 184: The use of the word "concerned" is unclear. Should this be "discerned" or "determined"?
-Page 10 Line 187: patients undergoing spinal anesthesia for a cesarean section do NOT have controlled ventilation as you mention here.
-Page 10 Line 194: Please move your hypothesis to the introduction.
-Page 10 Line 198: Here you state patients are "undergoing intravenous anesthesia for cesarean delivery". Your written inclusion criteria was for general anesthesia and many other portions of the paper describe spinal anesthesia. Please clarify this throughout the paper.
-Page 11 Line 201: "showed" should be replaced with "demonstrated"
-Page 11, Line 222: was esketamine considered to be included in this study?
-Many of the papers included in review were not targeted at using ketamine for PDPH symptomatic relief. It is difficult for me to believe that you are able to determine if ketamine is beneficial for raising ICP to improve PDPH symptoms if the studies used were looking at ketamine to improve shivering or for postoperative analgesia. Nausea/vomiting from the sample found could be due to a number of issues, not just PDPH. It is difficult for me to believe the data being evaluated is completely relevant to the research question you are positing.
Author Response
Reviewer #2: I would like to congratulate the authors on this paper evaluating the impact of ketamine on side effects following spinal anesthesia. Overall, the paper provides a good idea, but is difficult to understand at times. The incidence of PDPH following spinal anesthesia is very rare and the sample size reviewed in this paper are likely not powered to determine if ketamine impacted this. Was the use of ketamine for PDPH considered as the primary search? These patients with known headache or inadvertent dural puncture during epidural placement are likely more likely to have changes in symptoms with ketamine treatment compared to those who underwent spinal anesthesia alone for cesearean delivery.
Response: We are grateful for these comments which helped us to better describe the rationale of our study. Meanwhile we thank the reviewer for pointing out a flaw in our study and agree with his suggestion. We would like to elaborate further on this point. We are interested in the effect of ketamine in raising intracranial pressure and believe that this effect may improve the problem of low cranial pressure headache after clinical intravertebral anesthesia. Therefore, I attempted to search for any relevant clinical RCT studies and only one study focused on assessing this effect of ketamine. Therefore, I began searching the literature for other cesarean procedures performed under intralesional anesthesia involving symptoms related to low cranial pressure. These studies were eventually included in the current study.
I agree with the question posed by this reviewer as to whether the subjects experienced cerebrospinal fluid leakage resulting in hypocranial pressure. However, our study was more inclined to assess whether ketamine would be beneficial in such patients, namely, whether the intracranial pressure elevating effects of ketamine would alleviate symptoms associated with hypocranial pressure, such as headache, nausea, and vomiting, and whether these symptoms were caused by hypocranial pressure due to cerebrospinal fluid leakage or by the psychoactive effects of ketamine itself was not of undue concern. Of course, the effect of ketamine can be better demonstrated if it is clear that the patient has experienced intracranial pressure due to cerebrospinal fluid leakage.
Point-by-Point Response to the reviewers’ comments
- -The use of the term "vomit" throughout the manuscript should be replaced with "vomiting" for most of the instances.
Response: Thank you very much for your valuable comment. We have changed “voimt” to “vomiting” in the full text.
- -Page 1, Line 32: It is not clear to me what "progressive technology maturity" means.
Response: Our intention was to express that the gradual maturation of caesarean section technology is one of the reasons for the increase in the rate of caesarean section. We list the modified parts as follows: “The proportion of cesarean sections in deliveries is increasing due to gradual maturation of birth assistance techniques and social factors”.
- -Page 1, Line 32: The sentence doesn't read clearly. Consider removing "because it can not only" and replace with "Cesarean delivery [help pregnant women...]"
Response: We have corrected the sentence and list as follows: “Cesarean delivery can help pregnant women who were not allowed to be delivered normally due to problems like pelvic stenosis, uterine opening insufficiency, placenta previa, and also avoid the occurrence of pelvic floor injury, urinary incontinence and vaginal relaxation and other complications caused by natural labor”.
- -Page 2 line 59: you state you "screened 2099 patients" but do you mean 2099 manuscripts? If so, this should be included in the results. If not, please clarify what yoyu mean by screening patients.
Response: We would like to express that we finally including a total of 2099 patients in the study, thank you for pointing this out, we have corrected it in the manuscript and listed as follows: “We searched Medline, Embase, Web of Science, and Cochrane databases from 1976 to 2021 and, resulting in the inclusion of 2099 patients”.
- -Page 2 Line 63: your study aims to evaluate patients undergoing spinal anesthesia, yet your inclusion criteria includes patients undergoing surgery with general anesthesia. Please provide clarification. Also include anesthesia types to be excluded in the exclusion criteria and procedure types/anesthesia types to be included.
- Response: Thank you very much for your valuable comment. We apologize for the mistake here in the article, we did include patients under spinal anesthesia rather than general anesthesia for cesarean delivery. Thank you for your careful review and for pointing this out. We have corrected this error and revised the inclusion and exclusion criteria to ensure their accuracy,and we listed revised version as follows: “The Inclusion criteria were as follows: a. Randomized controlled trial, b. American Society of Anesthesiologists ( ASA ) Ⅰ-Ⅱ, c. Undergoing cesarean section with spinal anesthesia, d. participants treated intravenously with ketamine, e. Articles written in English. The exclusion criteria were as follows: a. Unpublished clinic trail; b. Unavaila-ble full texts; c. Control group was a drug other than normal saline”.
- -Were patients with known dural punctures included (such as inadvertent dural puncture during epidural placement)?
Response: Since the literature we included did not mention whether the patient had a dural puncture, we were also unable to determine whether the patient presented with dural puncture.
- -Page 2 Line 70: "Relevant Outcomes" and "Reported Results" shouldn't be capitalized. You can list the outcomes within parentheses after "relevant outcomes" and remove the subsequent sentence.
Response: Thank you very much for your helpful advice. We have made changes in accordance with your suggestions. We list the modified parts as follows: “first author's name, year of publication, number of patients, age, BMI (height and weight values were provided in some studies), ketamine dose, needle size, relevent outcomes (the incidence of perioperative headache, nausea, and vomiting) and re-ported results.”
- -Page 2 Line 73: This sentence and the subsequent sentences discussing characteristics of studies evaluated should be moved to the results section. Please include the steps for review of the identified studies, but don't include individual information.
Response: Thank you very much for your helpful advice. We have moved the relevant parts of the original discussion to the results section of the revised manuscript. At the same time we have added review steps as follows: “By searching the databases mentioned above, Two-hundred-eight studies were re-trieved in the systematic review. Eighty-seven duplicate studies were excluded. After screening of titles and abstracts, only thirty studies were considered eligible for a full reading. Twelve studies met all selection criteria”
- -The methods section should not refer to figures. The results/discussion section should reference figures.
Response: Thank you for your helpful advice. We have deleted the figures reference.
- -The use of the term "literatures" should be changed to "studies" or "manuscripts".
Response: We have replaced the word “literatures” in the text with the “studies”
- -Table 1: the use of "K/C" makes the control group appear as the denominator for the ketamine group. Please correct this and provide relevant denominators for each characteristic reported.
Response: Thank you very much for your helpful advice. We have changed "K/C" to “K;C”.
- -Table 1: Was information regarding spinal needle size or shape included in data collection? Both of these factors would contribute to the potential for PDPH.
Response: This is really an important information. We found the puncture needle size from the included studies and added the information to Table 1
- -Page 10 Line 156: It is not clear to me why you abandoned one study each time in the sensitivity analysis. What was this sensitivity analysis aiming to accomplish? Please add this to the methods section and provide further context within the results.
Response: Thank you very much for your valuable comment. The purpose of the sensitivity methods analysis was to explore the effect of excluding certain low-quality studies on the total effect. we have revised it and listed as follows:“A leave-one-out test, consisting of calculating the pooled risk ratio by sequentially ex-cluding one study, was performed to identify studies with a strong influence on the results.”
- -Page 10 Line 166. Was this paper limited to only emergency cesarean sections? This is the first mention of emergency cesarean delivery. Please reconsider the use of "emergency". Additionally, the spinal anesthetic is providing surgical anesthesia, not "analgesia".
Response: Thank you very much for your valuable comment. Our study was not limited to emergency cesarean delivery, but was only phrased as such for the purpose of introducing cesarean delivery in the discussion. To avoid misinterpretation of the article, we have corrected the expression accordingly and removing the word “emergency”. We also thank you for your correction regarding the role of spinal anesthesia in cesarean delivery, we have revised it and listed as follows: “Cesarean section is one of the most common operations in operating room, and the spinal anesthesia can provide a sufficient surgical anesthesia”.
- -Page 10 Line 174. Consider removing "decubitus", using the term "analgesic medications" instead of "analgesic sedation", and "epidural blood patch" instead of "epidural blood filling therapy. Additionally, I don't quite understand what "It is more important for patients to effectively prevent and alleviate their perioperative discomfort" means in this context.
Response: Thank you very much for your helpful advice. We have made the changes you suggested and listed here: “Studies have shown that hypocranial pressure syn-drome will affect the prognosis and recovery of pregnant women[31]. Although dif-ferent treatment modalities are available depending on the severity, including lying down, massive fluid administration, analgesic medications, and epidural blood patch[32, 33], however,effective prevention of PDPH is more necessary”.
- -Page 10 Lines 176-180: It is tough to follow the wording of these sections. Please make more clear.
Response: Thank you very much for your valuable comment. We have modified the sentence and the main intent of this paragraph is to show that it is reasonable to use headache, nausea and vomiting as indicators to assess the treatment of low cranial pressure. We have revised it and listed as follows: “Headache is a characteristic symptom of low intracranial pressure[29], most im-portantly postural headache, accompanied by nausea and vomiting[30]. Therefore,We regard the incidence of headache, nausea and vomiting as indicators to assess the treatment of hypocranial pressure”.
- -Page 10 Line 184: The use of the word "concerned" is unclear. Should this be "discerned" or "determined"?
Response: Thanks for your helpful advice. We have followed your suggestion to replace "concerned" with "discerned".
- -Page 10 Line 187: patients undergoing spinal anesthesia for a cesarean section do NOT have controlled ventilation as you mention here.
Response: Thank you very much for your valuable comment. Here we revised this sentence and listed as follows: “Therefore, the cesarean patients included in our study could ensure the intracranial pressure-raising effect of ketamine with spontaneous breathing.”
- -Page 10 Line 194: Please move your hypothesis to the introduction.
Response: We have moved the hypothesis to the introduction.
- -Page 10 Line 198: Here you state patients are "undergoing intravenous anesthesia for cesarean delivery". Your written inclusion criteria was for general anesthesia and many other portions of the paper describe spinal anesthesia. Please clarify this throughout the paper.
Response: Thank you very much for your valuable comment. We apologize for such an error in our article, for which we apologize, as we neglected the rigor of scientific research. We have made a correction here and in the inclusion criteria, and we state that all patients included in this study were cesarean section patients under spinal anesthesia.
- -Page 11 Line 201: "showed" should be replaced with "demonstrated".
Response: We have followed your suggestion to replace "showed" with " demonstrated ".
- -Page 11, Line 222: was esketamine considered to be included in this study?
Response: Esketamine was not addressed in our study. This is because it has been used in clinical practice for a shorter period of time compared to ketamine and we did not search for enough relevant studies to perform a meta-analysis. However, esketamine is now widely used in clinical practice with less psychoactive effects compared to ketamine. and it is more likely to reduce the incidence of pdph. This is just our outlook and vision, and we may consider clinical RCTs in this area in the future.
- -Many of the papers included in review were not targeted at using ketamine for PDPH symptomatic relief. It is difficult for me to believe that you are able to determine if ketamine is beneficial for raising ICP to improve PDPH symptoms if the studies used were looking at ketamine to improve shivering or for postoperative analgesia. Nausea/vomiting from the sample found could be due to a number of issues, not just PDPH. It is difficult for me to believe the data being evaluated is completely relevant to the research question you are positing.
Response: Thank you very much for your valuable comment. I agree with you that it would have been more convincing if this study had looked directly for studies in which ketamine improved symptoms associated with low cranial pressure, and we searched for evidence of this, but unfortunately there were few such studies. Therefore, we started looking for studies that included low cranial pressure-related symptoms as a side effect, and after screening, we included them in the study. As you mentioned, headache, nausea and vomiting may be multifactorial and not caused by hypocranial pressure, so the focus of our study was not whether the patient actually had hypocranial pressure, but whether the use of ketamine improved these associated symptoms, which are also psychoactive effects of ketamine. We were more interested in knowing whether ketamine actually reduced these symptoms in such patients.However,we acknowledge that this is a shortcoming in our study.
